# ROBUST HETEROGENEOUS TREATMENT EFFECT ESTIMATION UNDER COVARIATE PERTURBATION

## ABSTRACT

Heterogeneous treatment effect estimation has important applications in fields such as healthcare, economics, and education, attracting increasing attention from both the academic and industrial community. However, due to the lack of robustness against perturbation on the covariates, most existing causal machine learning methods may not perform well in practice in treatment effect estimation. In this paper, we mitigate this problem using the idea of adversarial machine learning. We first show that our loss of interest, the adversarial loss, is partly bounded by the Lipschitz constant of the causal prediction model. Next, we propose a representation learning-based **R**obust **H**eterogeneous **T**reatment **E**ffect (RHTE) framework which estimates heterogeneous treatment effect under covariate perturbation by controlling the empirical loss, Lipschitz constant, and distance metric simultaneously. Theories are derived to guarantee the performance and robustness of our estimation. To the best of our knowledge, this is the first work proposing robust causal representation learning methods under covariate perturbation. Extensive experiments on both synthetic examples and standard benchmarks demonstrate the effectiveness and generality of our RHTE framework.

## 1 INTRODUCTION

Treatment effect measures the causal impact of a treatment or intervention on a targeted outcome. Identifying and estimating treatment effect is of great significance in observational studies across domains, such as healthcare (Shalit, 2020), computer vision (Santurkar et al., 2019; Elsayed et al., 2018) and recommender system (Wang et al., 2021; 2022). In some scenarios, our estimand of interest is not the average treatment effect (ATE) on the entire population, but the one on specific subgroups of individuals with the same covariates, e.g. age, gender, etc., which is generally referred to as **Conditional Average Treatment Effect (CATE)** or **Heterogeneous Treatment Effect (HTE)** (Wager & Athey, 2018; Jacob, 2021; Fan et al., 2022).

Due to the prevailing existence of confounders that influence both the treatment and outcome variables, the distributions of treated and control groups are often imbalanced, posing challenges in obtaining accurate estimation for CATE. From the causal machine learning community, Counterfactual Representation Learning Johansson et al. (2016) is proposed to address the challenge by learning a balanced representation between treatment and control groups from the covariates through distance metrics, while minimizing causal effect estimation error. CFR (Shalit et al., 2017) method, together with other variants such Dragonnet (Shi et al., 2019) and TARNet (Shalit et al., 2017) are shown to have solid theoretical guarantees and perform well in many real-world settings.

In many observational studies, however, the variables including treatment $T$, outcome $Y$, and covariates $X$ may not be accurately measured. For example, when estimating the conditional average treatment effect on cardiovascular disease, the Electrocardiogram (ECG) data is a perturbed observation of the heart status due to the limitation of scan resolution. Scans with low resolution could result in wrong treatment evaluations and decisions made by doctors with unpredictable consequences. Since the measurement error occurs on the covariates in the above example, we refer to such measurement error as covariate perturbation. Some works have discussed measurement error in causal inference (Imai & Yamamoto, 2010; Pearl, 2012; Kuroki & Pearl, 2014), addressing measurement error in outcome (Shi et al., 2019), treatment (Xiao et al., 2019; Zhu et al., 2022) or covariate (Kallus et al., 2018; Shu & Yi, 2019). However, the above methods require strong

assumptions for either extra variables (instrumental variables, proxy variables, etc.) or distribution family, e.g. exponential family, to construct unbiased estimators for the treatment effect. These assumptions cannot be tested from observed data, and restrict the generality of the methods.

To fill this gap, in this paper, we propose a novel and effective framework called **R**obust **H**eterogeneous **T**reatment **E**ffect (RHTE) to achieve robustness representation learning method under covariate perturbation. Under the potential outcome framework, we first formulate the problem using adversarial samples, and define our loss of interest, the Expected Adversarial Factual Loss, which controls the estimation error in the worst case. By constructing inequalities, we find that the adversarial loss can be bounded when we simultaneously control the Expected Factual Loss and Lipschitz constant of the model. This inspires us to design a representation learning framework that estimates CATE while controlling the Lipschitz constant and the representation distributions through the IPM metric. To constrain the Lipschitz constant, we propose two types of regularizations, called Orthonormality Regularization and RKHS Regularization, and prove their validity. Then, theoretical results are constructed to justify the robustness of RHTE estimation. Generalization bounds indicate that we can control the adversarial losses by taking into account the Lipschitz constant of the causal prediction model, along with empirical losses and the discrepancy between representation distributions. This implies we are able to control the error on adversarial samples, and hence on the real covariate, which proves the robustness of our estimation. Finally, we conduct experiments on both synthetic examples and standard benchmark datasets. Results show that RHTE outperforms baseline methods in most cases and makes robust estimations under covariate perturbation.

In summary, the main contributions of this paper can be concluded as follows:

- We address measurement error in causal machine learning methods, which is significant in application.
- We formalize the problem with the adversarial sample framework and control the adversarial loss through the Lipschitz constant, providing an approach to understanding covariate perturbation and measurement error.
- A robust estimation framework of CATE under covariate perturbation is proposed, with theories established to guarantee its performance and justify its robustness.
- Extensive experiments are conducted on both synthetic examples and standard benchmark datasets to demonstrate the effectiveness of our proposed method.

## 2 RELATED WORK

**Conditional average treatment effect estimation.** How to effectively and accurately estimate conditional average treatment effect has recently attracted increasing attention from the research community. It basically aims to discover the underlying patterns of the distribution between the treated and control groups. To model this character, early methods are based on re-weighting methods (Austin, 2011; Imai & Ratkovic, 2014; Fong et al., 2018) that is an effective approach to overcome the selection bias induced by the existence of covariates in observational studies. Other widely used techniques for CATE estimation belong to traditional machine learning, including Bayesian Additive Regression Trees (BART) (Hill, 2011), Random Forests (RF) (Breiman, 2001), Causal Forests (CF) (Wager & Athey, 2018), etc. These methods have more flexibility and predictive ability in balancing the distribution between treated and control groups compared to re-weighting methods. In addition, some promising works like S-Learner (Nie & Wager, 2021) and R-Learner (Künzel et al., 2019) are based on meta-learning to utilize any supervised learning or statistical regression methods to estimate the treatment effect. In recent years there have been plenty of studies adapting more sophisticated mechanisms to measure CATE. For example, DragonNet (Shi et al., 2019) design three-head components to predict the treatment effects as well as adjust the distribution by a process of inferring treatments. Besides, more cutting-edge mechanisms like Integral Probability Metric (IPM) (Qin et al., 2021; Johansson et al., 2016; Wu et al., 2022; Wang et al., 2023) are applied to minimize generalization bound for treatment effect estimation, which is composed of factual loss and the discrepancy between the treated and control distributions. The representative CFR (Shalit et al., 2017) method enforces the similarity between the distributions of treated and control groups in the representation space by a penalty term IPM, (Demirel et al., 2024) use the additional observational study to supplement the randomized clinical trial data, (Guo et al., 2024; Yan et al.) employ Meta-analysis and Optimal Transport to measure the inverse propensity score and (Li et al.) present a

generative approach to align the target population, and can be able to reduce the distribution shifts between treated and control groups.

While the boundary of estimation of CATE from observational data has been pushed by these models, an important problem is still under-explored, that is the robustness of the treatment effect predicted by deep neural networks when their input is subject to an adversarial perturbation. In this paper, we bridge this gap by proposing two types of regularizations called Lipschitz regularization and RKHS regularization to the original causal models for encouraging smoothness as well as improving the generalization performance.

**Adversarial machine learning.** Adversarial machine learning refers to techniques against adversarial perturbations (Huang et al., 2011). In the past few years, in order to facilitate the security and robustness of a model, adversarial machine learning has been widely applied to the machine learning community. For example, Cisse et al. (2017); Virmaux & Scaman (2018); Zhang et al. (2021) incorporated some adversarial examples or robustness regularization into the original objective for tackling sensitive issues in neural networks. In addition to that, some works (Deldjoo et al., 2021; Tian & Xu, 2021) attempt to enhance the robustness of the recommender system and audio-visual learning model respectively, and simultaneously improve its generalization performance via a way of adversarial optimization framework. Another important application is in computer vision (Santurkar et al., 2019; Elsayed et al., 2018), in which the adversarial examples are used to enhance the parameters of the original model. Nonetheless, to the best of our knowledge, we are the first work that integrate adversarial machine learning techniques into causal inference for CATE estimation. More importantly, we provide theoretical analysis on the expected precision in the estimation of heterogeneous effect (PEHE) loss and design two types of regularizations for encouraging robustness.

## 3 PROBLEM SETUP

We formalize our problems under the Neyman-Rubin potential outcomes framework as follows (Rubin, 2005). Consider an observational study in which each unit receives a binary treatment $T \in \mathcal{T} = \{0, 1\}$. Let $X \in \mathcal{X} \subset \mathcal{R}^d$ be the covariate in a bounded subset of $\mathcal{R}^d$, $Y \in \mathcal{Y} \subset \mathcal{R}$ be the observed outcome and the bounded outcome space, and $Y_0, Y_1$ be the potential outcome under treatment $T = 0$ and $T = 1$. In this paper, we mainly focus on estimating the conditional average treatment effect (CATE) (Shalit et al., 2017):

$$\tau(x) := \mathbb{E}[Y_1 - Y_0 | X = x]. \tag{1}$$

The fundamental problem in estimating CATE is that for any unit we only have one observed outcome. Therefore, it is hard to make inferences on both potential outcomes $Y_0$ and $Y_1$. In order to identify and estimate the effect above, we assume Stable Unit Treatment Value Assumption (SUTVA) as well as the following classical assumptions in causal inference hold (Yao et al., 2021):

**Assumption 1** (Consistency). *The observed outcome equals to the potential outcome under assigned treatment, e.g. $(Y|T = t) = Y_t$ for any unit and $t \in \mathcal{T}$.*

**Assumption 2** (Strong Ignorability). *$(Y_0, Y_1) \perp\!\!\!\perp T|X$ with $0 < p(T = 0|X) < 1$ for $\forall X \in \mathcal{X}$.*

Under the above assumptions, CATE can be identified as

$$\tau(x) = \mathbb{E}[Y|X = x, T = 1] - \mathbb{E}[Y|X = x, T = 0], \tag{2}$$

and the estimation problem turns into building up models for the conditional outcome $E[Y|X, T]$. Representation learning builds the conditional outcome model $E[Y|X = x, T = t] = f(\Phi(x), t)$ by finding a one-to-one representation function $\Phi : \mathcal{X} \to \mathcal{R}^l$. Let $L : \mathcal{Y} \times \mathcal{R}^l \to \mathcal{R}$ be the loss function. The model is trained by minimizing $L(y, f(\Phi(x), t))$, while balancing distributions $p_\Phi^{t=1} := p(\Phi(x)|t = 1)$ and $p_\Phi^{t=0} := p(\Phi(x)|t = 0)$ through a Integral Probability Metric (IPM) distance $\text{IPM}_G(p, q) := \sup_{g \in G} \left| \int_{\mathcal{S}} g(s)(p(s) - q(s))ds \right|$, where $G$ is the function class scaled expected loss lies in. For common function families $G$, IPM is a true metric over the corresponding set of probabilities (Shalit et al., 2017; Qin et al., 2021). When we let $G$ be the family of 1-Lipschitz functions, i.e., $G = \{g : ||g||_{Lip} \le 1\}$ we obtain the Wasserstein distance denoted by $WASS_G(\cdot, \cdot)$ between distributions. When $\mathcal{H}$ represents a Reproducing Kernel Hilbert Space (RKHS) (Sriperumbudur et al., 2009), and our function class is $G = \{g \in \mathcal{H} \ s.t. \ ||g||_{\mathcal{H}} \le 1\}$, IPM metric turns out to be the Maximum Mean Discrepancy denoted by $MMD_G(\cdot, \cdot)$.

## 4 ESTIMATION AND THEORIES

The performance of the representation learning method is justified through the expected Precision in Estimation of Heterogeneous Effect (PEHE) (Hill, 2011) loss on $f$:

$$\epsilon_{PEHE}(f) = \int_{\mathcal{X}} (\hat{\tau}(x) - \tau(x))^2 p(x) dx. \tag{3}$$

While $\epsilon_{PEHE}(f)$ measures the error between estimated and real CATE $\hat{\tau}(x)$ and $\tau(x)$, an underlying assumption is that covariate $X$ has been accurately observed. In many practical settings, however, the observed covariate is actually a perturbed observation of the real covariate $X_r$, e.g. $X = X_r + \delta_{X_r}$, where $\delta_{X_r}$ is a perturbed term. Using data suffering from severe covariate perturbation would result in predicting incorrect treatment effects with high confidence. In this section, we will propose methods and derive theories to find an estimation of CATE $\hat{\tau}(\tilde{x})$ using $\tilde{x}$, and derive theories to ensure its robust performance under covariate perturbation.

### 4.1 ADVERSARIAL SAMPLE AND LOSS

To guarantee the robustness of model performance under covariate perturbation, we aim at bounding its loss in the worst case when we estimate the effect using the adversarial sample. In this paper, we define and study the following spherical perturbation:

**Definition 1** (Spherical Perturbation). For a metric $|| \cdot ||$ in $\mathcal{X}$, there exists $\epsilon > 0$ such that $X_r$ lies in an $\epsilon$-ball centered at $X$, i.e. $||X - X_r|| < \epsilon$ for any $X \in \mathcal{X}$. $\epsilon$ is called the level of perturbation.

The condition of spherical perturbation in common cases, for example, when $\delta_{x_r} \sim N(0, \sigma^2)$. In this case, we can use $\ell_p$ norm in $\mathcal{X}$, set $p$ to be any even number greater than 0, and $\epsilon = [(p-1)!!]^{1/p}\sigma$. Under the above assumption, the adversarial sample of a unit with $X = x$ is formally defined as:

$$x_{adv} = \arg\max_{||\tilde{x}-x|| \leq \epsilon} L(f(\Phi(\tilde{x}), t), y). \tag{4}$$

The adversarial sample represents the worst sample with maximal loss in the area $X_r$ possibly lies. Since the only thing we know is that $X_r$ is in the $\epsilon$-ball, we can control the model performance in the entire ball only through its loss over the adversarial sample. For this sake, we aim at controlling the following Expected Adversarial Factual Loss $\epsilon_{Fadv}$:

**Definition 2.** For the adversarial examples, the expected adversarial factual loss of $f$ and $\Phi$ is

$$\epsilon_{Fadv}(f, \Phi, \epsilon) = \int_{\mathcal{X} \times \mathcal{T} \times \mathcal{Y}} L(y, f(\Phi(x_{adv}), t)) p(x, t, y) dx dt dy. \tag{5}$$

Note that this is different from the Expected Factual Loss generally studied in representation learning methods in that it computes the loss using the adversarial sample $x_{adv}$ rather than $x$:

$$\epsilon_F(f, \Phi, \epsilon) = \int_{\mathcal{X} \times \mathcal{T} \times \mathcal{Y}} L(y, f(\Phi(x), t)) p(x, t, y) dx dt dy. \tag{6}$$

The following lemma shows the relation between $\epsilon_{Fadv}$ and $\epsilon_F$.

**Lemma 1.** *Let $\epsilon$ denote the level of the perturbation. Assume that $L(y, f(\Phi(x), t))$ is a Lipschitz function with regard to $f$, with $\lambda_L$ being the Lipschitz constant. Assume that $f(\Phi(x), t)$ is a Lipschitz function with regard to $x$, and $\Lambda_f$ stands for the Lipschitz constant. Then we have*

$$\epsilon_F(f, \Phi) \leq \epsilon_{Fadv}(f, \Phi) \leq \epsilon_F(f, \Phi) + \lambda_L \Lambda_f \epsilon.$$

*Proof.* From the Lipschitz condition of $L$ and $f$, for any $\tilde{x}$ and $x \in \mathcal{X}$, we have

$$|L(y, f(\Phi(\tilde{x}), t)) - L(y, f(\Phi(x), t))| \leq \lambda_L |f(\Phi(\tilde{x}), t) - f(\Phi(x), t))| \leq \lambda_L \Lambda_f \epsilon$$

Therefore,

$$\epsilon_{Fadv} \leq \epsilon_F + |\epsilon_{Fadv} - \epsilon_F|$$

$$\leq \epsilon_F + \int_{\mathcal{X} \times \mathcal{T} \times \mathcal{Y}} \max_{||\tilde{x}-x|| \leq \epsilon} |L(y, f(\Phi(\tilde{x}), t)) - L(y, f(\Phi(x), t))| \, p(x, t, y) dx dt dy$$

$$\leq \epsilon_F + \lambda_L \Lambda_f \epsilon$$

$\square$

*Remark.* The Lipschitz constant for a function $g : \mathcal{M} \to \mathcal{N}$ is defined as $||g||_{Lip} = \sup_{x,y \in \mathcal{M}} \frac{||g(x)-g(y)||_{\mathcal{N}}}{||x-y||_{\mathcal{M}}}$, where $|| \cdot ||_{\mathcal{M}}$ and $|| \cdot ||_{\mathcal{N}}$ means the norm in each space.

Lemma 1 shows $\epsilon_{Fadv}$ is greater than $\epsilon_F$ itself, while can be upper-bounded by the sum of $\epsilon_F$ and multiplication of Lipschitz constants and level of perturbation. Note that $\lambda_L$ only depends on the loss we choose. Therefore, training model $f$ only affects $\Lambda_f$. Lemma 1 provides us insights that in order to control $\epsilon_{Fadv}$, we have to control $\epsilon_F$ and Lipschitz constants simultaneously. This inspires the estimation method in the next section.

## 4.2 ESTIMATION

We estimate model $f$ and representation function $\Phi$ through optimizing the following equation:

$$\min_{f,\Phi} \ \frac{1}{m} \sum_{i=1}^{m} w_i \cdot L(y_i, f(\Phi(x_i), t_i)) + \beta \cdot \Re(f) + \alpha \cdot \text{IPM}_G(\hat{p}_\Phi^{t=1}, \hat{p}_\Phi^{t=0}),$$

$$s.t \ \ w_i = \frac{t_i}{2u} + \frac{1-t_i}{2(1-u)}, \ \ \text{where} \ \ u = \frac{1}{m} \sum_{i=1}^{m} t_i. \tag{7}$$

The weights $w_i$ balances the difference between the sizes of treatment and control group (Shalit et al., 2017), $\hat{p}_\Phi^{t=1}$ and $\hat{p}_\Phi^{t=0}$ are empirical distribution of $p_\Phi^{t=1}$ and $p_\Phi^{t=0}$ respectively, and recall that $\text{IPM}_G(\cdot, \cdot)$ is the distance metric between these two distributions. We use two specific types of IPM, WASS, and MMD, with details provided in the experiment part. $\Re(f)$ is a Lipschitz regularization term with details discussed later. Through the estimation above, we minimize the empirical loss of $L(y, f(\Phi(x), t))$ while balancing the empirical distributions of representations in treatment and control groups, which helps us control the expected factual loss $\epsilon_F$. Meanwhile, we control the Lipschitz constants of $f$ through regularizing over $\Re(f)$. Consequently, we are able to control $\epsilon_{Fadv}$, which will be discussed in Theorem 1. Besides, we can also derive a generalization bound for the adversarial version of PEHE, which further encourages its robustness, see Theorem 2 for details.

The choice of regularization term $\Re(f)$ depends on the norm in $\mathcal{X}$ and the functional space of $f_{\Phi,t}$. Next, we propose two kinds of Lipschitz regularization terms $\Re(f)$ to bound the Lipschitz constant.

**Orthonormality Regularization**

In this paper, representation function $\Phi(x)$ is estimated through an $l_\Phi$-layer feed-forward neural network, and outcome model $f(r, t)$ is an $l_t$-layer network with regard to $r$. Let $W_\Phi^k$ and $W_t^k$ be the weight matrix for the k-th layer of the network for $\Phi(x)$ and $f(r, t)$, respectively.

Consider the $\ell_2$-norm in $\mathcal{X}$. For a layer with weight matrix $W : \mathcal{R}^{n_0} \to \mathcal{R}^{n_1}$, we have

$$||Wx - W\tilde{x}||_2 \le ||W||_2 \cdot ||x - \tilde{x}||_2, \tag{8}$$

for any $x, \tilde{x} \in \mathbb{R}^{n_0}$, where $||W||_2$ is the spectral norm of matrix. Therefore, the Lipschitz constant for this layer can be bounded by $||W||_2$. Since $f_{\Phi,t}(x)$ is a two-branch neural network with shared layers on $\Phi(x)$, applying the composition rules in estimating the Lipschitz constants (Tsuzuku et al., 2018), the Lipschitz constant of $f_{\Phi,t}(x)$ with regard to $x$ denoted by $\Lambda_f$ can be bounded by production of spectral norms as follows:

$$\Lambda_f \le \prod_{k=1}^{l_\Phi} ||W_\Phi^k||_2 \cdot \max\{\prod_{m=1}^{l_1} ||W_1^m||_2, \prod_{m=1}^{l_0} ||W_0^m||_2\}. \tag{9}$$

The works in parseval tightness theory (Kovačević et al., 2008; Cisse et al., 2017) demonstrate that the orthonormality of weight matrices is sufficient to control the spectral norm. Following the above idea, we aim to constrain the parameters with orthonormality for each transformation layer through

$$\Re_t^k(f) = \frac{1}{2}||W_t^{k^T} W_t^k - I||_2^2, \tag{10}$$

and $\Re_\Phi^k(f)$ correspondingly, where $I$ refers to the identity matrix. The gradient of this regularization term is $\nabla_{W_t^k} \Re_t^k(f) = (W_t^k W_t^{k^T} - I)W_t^k$. And the regularization term $\Re(f)$ is constructed by

$$\Re(f) = \sum_{k=1}^{l_\Phi} \Re_\Phi^k(f) + \sum_{m=1}^{l_0} \Re_0^m(f) + \sum_{m=1}^{l_1} \Re_1^m(f). \tag{11}$$

The regularization above helps us constrain the Lipschitz constant $\Lambda_f$. In the extreme case when $\Re(f) = 0$, all $\Re_\Phi^k(f)$ and $\Re_t^m(f)$ equals 0, indicating that the weight matrix $W$ for each layer is orthogonal and $||W||_2 = 1$. Therefore, from Eq. (9) the Lipschitz constraint is bounded by $\Lambda_f \leq 1$.

**RKHS Regularization**

Assume $f_{\Phi,t}(x)$ lies in a reproducing Hilbert kernel space $\mathcal{H}$ (Sriperumbudur et al., 2009), and denote the norm and reproducing kernel function as $||\cdot||_\mathcal{H}$ and $K(\cdot,\cdot)$, respectively. Define the norm on $\mathcal{X}$ as $||x - y|| = ||K(.,x) - K(.,y)||_\mathcal{H}$. From reproducing property, we have

$$||f_{\Phi,t}(x) - f_{\Phi,t}(y)|| = \langle f_{\Phi,t}(\cdot), K(\cdot,x) - K(\cdot,y) \rangle \leq ||f_{\Phi,t}||_\mathcal{H} \cdot ||x - y||. \tag{12}$$

Therefore, we have $\Lambda_f \leq ||f_{\Phi,t}||_\mathcal{H}$, which controls Lipschitz constant $\Lambda_f$ through the RKHS norm. From such bound, we can construct $\Re(f)$ to constrain the RKHS norm to control $\Lambda_f$:

$$\Re(f) = ||f_{\Phi,t}||_\mathcal{H} - 1. \tag{13}$$

When $\Re(f) = 0$, $\Lambda_f \leq ||f_{\Phi,t}||_\mathcal{H} = 1$, which bounds the Lipschitz constant of $f$.

### 4.3 THEORETICAL RESULTS

In this section, we will list theoretical results which guarantee the robust performance of our estimation under covariate perturbation. The complete proofs and details are presented in the Appendix. To begin with, recall that $\epsilon$ is the level of perturbation. Let $D = \{(x_i, t_i, y_i)\}_{i=1}^m$ denote the training data drawn from the sample space $\mathcal{D} = \mathcal{X} \times \mathcal{T} \times \mathcal{Y}$, and let $\mathcal{D}_t$ be its subspace $\mathcal{D}_t = \mathcal{X} \times \{t\} \times \mathcal{Y}$. First, we derive the following bound for expected adversarial factual loss $\epsilon_{Fadv}$ in Definition 2:

**Theorem 1.** *Let $C_t(\mathcal{D}, \epsilon)$ be the covering number of $\mathcal{D}_t$ using $\epsilon$-balls, $C_p(\mathcal{D}, \epsilon) = max\{C_0(\mathcal{D}, \epsilon), C_1(\mathcal{D}, \epsilon)\}$, and $C_d = \sup_{x,t,W,y} L(y, f(\Phi(x), t))$, where $W$ is the parameter set of $f$ and $\Phi$. For $\delta > 0$, with probability at least $1 - \delta$ over the i.i.d. samples $\{(x_i, t_i, y_i)\}_{i=1}^m$, we have*

$$\epsilon_{Fadv}(f, \Phi) \leq \frac{1}{m} \sum_{i=1}^m L(y_i, f(\Phi(x_i), t_i)) + 2\lambda_l \Lambda_f \epsilon + C_d \sqrt{\frac{2C_p(\mathcal{D}, \epsilon)\ln 2 + 2\ln(1/\delta)}{m}}.$$

*Remark.* Theorem 1 provides an upper bound for the expected adversarial factual loss that controls the expected treatment effect estimation error caused by adversarial samples over factual distribution, which rationalizes the control on Lipschitz constant in order to bound $\epsilon_{Fadv}$.

Next, analogous to the PEHE loss commonly used to measure the performance of CATE estimation, under covariate perturbation, we define Adversarial PEHE loss as

$$\epsilon_{PEHEadv}(f) = \int_\mathcal{X} (\hat{\tau}(x_{adv}) - \tau(x_{adv}))^2 p(x)dx, \tag{14}$$

where $x_{adv}$ is the adversarial sample defined in (4). Adversarial PEHE loss helps us control the expectation of maximal square error caused by estimating using real covariate $x_r$. Estimation with low Adversarial PEHE loss indicates good generalization performance not only using the observed covariate to estimate CATE but also using real covariates, contained within the $\epsilon$-ball from the observation, to estimate CATE. We have the following bound on the Adversarial PEHE loss:

**Theorem 2.** *Let $G$ be a family of functions $g : \mathcal{R} \to \mathcal{Y}$. Assume that there exists an $\ell_2$ loss, $L : \mathcal{Y} \times \mathcal{Y} \to \mathcal{R}_+$, and a constant $C_\Phi > 0$, such that for fixed $t \in \{0, 1\}$, the per-unit expected adversarial loss function $\tilde{\ell}_{f,\Phi}(x, t) = \int_\mathcal{Y} L(Y_t, f(\Phi(x_{adv}), t))p(Y_t|x)dY_t$ obey $\frac{1}{C_\Phi} \cdot \tilde{\ell}_{f,\Phi}(x, t) \in G$. Let $C_Y$ be the minimum variance of the outcomes $Y_t$ under factual and counterfactual distributions. Then, with probability at least $1 - \delta$,*

$$\epsilon_{PEHEadv}(f, \Phi) \leq \frac{4}{m} \sum_{i=1}^m L(y_i, f(\Phi(x_i), t_i))$$

$$+ 4\left(\lambda_l \Lambda_f \epsilon + C_d \sqrt{\frac{2C_p(\mathcal{D}, \epsilon)\ln 2 + 2\ln(1/\delta)}{m}}\right) + 2\left(C_\Phi \cdot IPM_G(p_\Phi^{t=1}, p_\Phi^{t=0}) - C_Y\right).$$

*Remark.* Theorem 2 provides us insights that through controlling the empirical loss, Lipschitz constant, and distance between representation distributions simultaneously in (7), the generalized robust performance of RHTE estimator is guaranteed through the Adversarial PEHE loss.

## 5 EXPERIMENTS

### 5.1 EXPERIMENTAL SETUP

**Datasets.** CATE estimation is more difficult compared to prediction tasks since we rarely have access to ground-truth treatment effects in real-world scenarios. To measure the effectiveness of the proposed methods, we conduct extensive experiments based on two standard benchmark datasets, **ACIC** (Dorie et al., 2019) and **IHDP** (Hill, 2011), and two synthetic Multimodal datasets , **UTK-sim** and **TC-sim**. The ACIC dataset is a common benchmark dataset introduced by Dorie et al. (2019). It comprises 4,802 units (28% treated, 72% control) and 82 covariates measuring aspects of the linked birth and infant death data (LBIDD). The datasets are generated randomly according to the data-generating process setting. The IHDP dataset was based on the Infant Health and Development Program. It presented a semi-synthetic dataset for estimating causal effects. The covariates were created through a randomized experiment examining the impact of home visits by specialists on future cognitive scores. It consists of 747 units(19% treated, 81% control ) and 25 covariates measuring the children and their mothers. The UTK-sim dataset is generated from the combining of tabular data and UTK images Zhang et al. (2017), in which it consists of 1000 units (49% treated, 50% control), 2710 covariates representing the unit's profiles in images. The more details of generation process can refer to Deshpande et al. (2022). The TC-sim dataset is followed by Wang & Culotta (2020) where it consists of 3240 units (25% treated, 75% control), and 3071 covariates measuring toxic comment.

**Baselines.** We compare our model with the following 11 representative baselines: Tree-based methods Random Forests (RF) (Breiman, 2001) and Causal Forests (CF) (Wager & Athey, 2018), meta learning methods S-Learner (Nie & Wager, 2021) and T-Learner (Künzel et al., 2019), Balancing Neural Network (BNN) (Johansson et al., 2016), DragonNet (Shi et al., 2019), Treatment-Agnostic Representation Network (TARNet) (Shalit et al., 2017) as well as Counterfactual Regression with the Wasserstein metric (CFR$_{\text{WASS}}$) (Shalit et al., 2017) and the squared linear MMD metric (CFR$_{\text{MMD}}$) (Shalit et al., 2017), along with two extensions of CRF method Decomposed Representations for CounterFactual Regression (DeRCFR) (Wu et al., 2022), and Optimal Transport for Treatment Effect Estimation (ESCFR) (Wang et al., 2023).

**Experimental Details.** Our methods are implemented with BNN introduced by Johansson et al. (2016). For all four datasets, the network architecture is shared and the same set of hyperparameters is adopted. We set both hyperparameters $\alpha$ and $\beta$ to 1 except for the ablation study. More implementation details are provided in the Appendix.

Following the settings of previous studies (Shalit et al., 2017; Wu et al., 2022; Wang et al., 2023), we resent within-sample and out-of-sample results that are calculated on the training and test set respectively. The commonly used metric including Rooted Precision in Estimation of Heterogeneous Effect (PEHE) (Hill, 2011) is applied for evaluating the quality of treatment effects. Formally, they are defined as: $\sqrt{\epsilon_{\text{PEHE}}} = \sqrt{\frac{1}{n}\sum_{i=1}^{n}\left(\hat{\tau}_i - \tau_i\right)^2}$, where $\hat{\tau}_i$ and $\tau_i$ stand for the predicted CATE and the ground truth CATE for the $i$-th instance respectively.

In comparison with the 11 baselines mentioned above, we add extra perturbation to the test sets. More concretely, for given a test data point $x$, we generate a new covariate $x' = x + \delta_x$ to substitute for the original one. We choose level of noise $\delta_x$ in $\{\mathbb{U}(-0.1, 0.1)^{dim(x)}\}$.

### 5.2 EXPERIMENTAL RESULTS

**CATE Estimation.** The overall comparison results for four datasets with perturbation are presented in Table 1, from which we can see that compared to the standard benchmark datasets, the performance of all the models are a little higher on the synthetic datasets, which is because of the imbalanced distribution nature between treated and control groups , and verifies the difficulties of the treatment effects estimation task itself. Representation learning methods like DragonNet can usually obtain better performance than the traditional machine learning method like RF, which agrees with the previous works (Qin et al., 2021; Shalit et al., 2017), and verifies the usefulness of predicting the CATE by a deep neural network. Among representation learning models, the best performance is usually achieved when the model is based on the IPM distance metric. This is as expected since the

Table 1: Treatment effects estimation. In each module, we present each of the results with form mean ± standard deviation and we use bold fonts to label the best performance. Lower is better.

| Datasets | ACIC | | IHDP | | UTK-sim | | TC-sim | |
|---|---|---|---|---|---|---|---|---|
| Task | In-sample | Out-sample | In-sample | Out-sample | In-sample | Out-sample | In-sample | Out-sample |
| R.Forest | $4.05 \pm 1.36$ | $4.05 \pm 1.38$ | $6.29 \pm 9.48$ | $5.91 \pm 8.9$ | $0.33 \pm 0.01$ | $0.34 \pm 0.02$ | $0.89 \pm 0.52$ | $0.88 \pm 0.52$ |
| C.Forest | $1.88 \pm 0.76$ | $1.89 \pm 0.78$ | $4.94 \pm 7.63$ | $4.91 \pm 7.48$ | $0.25 \pm 0.01$ | $0.24 \pm 0.02$ | $0.86 \pm 0.44$ | $0.84 \pm 0.46$ |
| S-Learner | $3.83 \pm 1.42$ | $3.85 \pm 1.46$ | $6.27 \pm 9.39$ | $6.25 \pm 9.55$ | $0.27 \pm 0.01$ | $0.26 \pm 0.02$ | $0.94 \pm 0.25$ | $0.88 \pm 0.32$ |
| T-Learner | $2.38 \pm 0.88$ | $2.44 \pm 0.88$ | $5.47 \pm 10.19$ | $5.58 \pm 10.55$ | $0.34 \pm 0.13$ | $0.35 \pm 0.14$ | $1.06 \pm 0.33$ | $0.97 \pm 0.37$ |
| BNN | $5.59 \pm 1.56$ | $5.57 \pm 1.54$ | $8.55 \pm 8.75$ | $8.4 \pm 8.52$ | $0.26 \pm 0.01$ | $0.27 \pm 0.01$ | $0.89 \pm 0.01$ | $\mathbf{0.82 \pm 0.08}$ |
| DragonNet | $1.78 \pm 0.44$ | $1.79 \pm 0.43$ | $2.54 \pm 3.09$ | $2.54 \pm 3.15$ | $0.19 \pm 0.02$ | $0.21 \pm 0.03$ | $0.89 \pm 0.39$ | $0.85 \pm 0.40$ |
| TARNet | $1.75 \pm 0.53$ | $1.80 \pm 0.56$ | $2.35 \pm 2.87$ | $2.4 \pm 2.85$ | $0.13 \pm 0.01$ | $0.14 \pm 0.02$ | $0.85 \pm 0.47$ | $0.83 \pm 0.5$ |
| CFR$_{MMD}$ | $1.71 \pm 0.4$ | $1.74 \pm 0.41$ | $2.28 \pm 2.67$ | $2.21 \pm 2.31$ | $0.20 \pm 0.02$ | $0.21 \pm 0.02$ | $0.90 \pm 0.52$ | $0.88 \pm 0.55$ |
| CFR$_{WASS}$ | $1.74 \pm 0.43$ | $1.78 \pm 0.47$ | $2.21 \pm 2.81$ | $2.22 \pm 2.65$ | $0.17 \pm 0.02$ | $0.17 \pm 0.02$ | $0.89 \pm 0.49$ | $0.86 \pm 0.52$ |
| DeRCFR | $1.79 \pm 0.49$ | $1.83 \pm 0.51$ | $3.23 \pm 4.62$ | $3.24 \pm 4.68$ | $0.18 \pm 0.03$ | $0.19 \pm 0.04$ | $0.89 \pm 0.10$ | $0.82 \pm 0.18$ |
| ESCFR | $2.73 \pm 1.1$ | $2.81 \pm 1.15$ | $3.84 \pm 5.39$ | $4.1 \pm 5.73$ | $0.21 \pm 0.03$ | $0.23 \pm 0.02$ | $\mathbf{0.85 \pm 0.43}$ | $0.84 \pm 0.44$ |
| RHTE$_{MMD}$ | $1.31 \pm 0.33$ | $1.33 \pm 0.34$ | $\mathbf{1.97 \pm 2.66}$ | $\mathbf{1.99 \pm 2.54}$ | $0.27 \pm 0.05$ | $0.28 \pm 0.06$ | $0.87 \pm 0.49$ | $0.86 \pm 0.5$ |
| RHTE$_{WASS}$ | $\mathbf{1.26 \pm 0.42}$ | $\mathbf{1.28 \pm 0.43}$ | $2.13 \pm 2.94$ | $2.12 \pm 2.79$ | $\mathbf{0.11 \pm 0.01}$ | $\mathbf{0.11 \pm 0.01}$ | $0.91 \pm 0.51$ | $0.88 \pm 0.53$ |

IPM distance metric based on the studied representation can effectively reduce the distribution shift between treated and control groups, improving the generalization performance of CATE estimation.

Encouragingly, our model can achieve the best performance on the most tasks across different datasets, where the improvements are mostly significant. The result is consistent with our theoretical analysis in section 3. Compared to the baselines, we introduce the Lipschitz regularization and RKHS regularization separately to reduce the Lipschitz constant of the treatment effects model, improving the robustness of treatment effects estimation. Between the different implementations of the IPM distance metric, we find that WASS is a little superior to MMD in most cases. We speculate that WASS is more suitable for balancing the representation distributions.

It is important to note that the effectiveness of our model on the synthetic Multimodal dataset UTK-sim validates that covariates perturbation may occur in Multimodal scenarios.

**Robustness Comparison.** To verify the effectiveness of the proposed two types of regularizations compared to simple adversarial defense-based methods, we also conduct experiments with baselines by simply combining three types of adversarial samples-based methods: Training-based methods, Architectural-based methods and Distillation-based methods Serban et al. (2020); Costa et al. (2024). In table 2, 'X(T)' 'X(A)',and 'X(D)' denote the training-based,architectural-based and distillation-based methods respectively when baseline is X. We apply our framework to ESCFR, TARNet, and DeRCFR, and use IHDP and UTK-sim as the experimental datasets. The conclusions on the other base models and datasets are similar and omitted. From the results presented in Table 2 we can see: among the baselines, the performance can be lower (In most cases) or better compared to the original baseline's performance presented in Table 1, we speculate that by simply combing adversarial defense methods to casual models can't address the problems that arise in causal inference,like distribution shift,etc. Additionally, in most cases, our methods can achieve the best performance compared to the baselines, and improvement is consistent on most datasets and evaluation metric. Above observations verify the effectiveness of the proposed two types of regularizations compared to simple adversarial defense methods.

**Effects of Varying Perturbation Level.** We further investigate our model with different levels of extra noise and compare it with ESCFR, TARNet, and DeRCFR on the datasets of IHDP and UTK-sim. More specifically, we add two new non-spherical types of perturbation $\delta_x$ on covariates. The first one is called Fast gradient sign method (FGSM),and the second one is called One-step target class method (OTCM). The detailed generation process could refer to section 4 in Puttagunta et al. (2023). The results are presented in Figure 1. By imposing small extra perturbation values on the input point, we can find that all of the performances on dataset IHDP and UTK-sim have been degraded jointly compared to Table 1 that added spherical types of perturbation. We speculate that the non-spherical perturbation could bring more noise than spherical perturbation in estimating HTE. It is encouraging to see that our framework can still outperform the base models in all task cases.

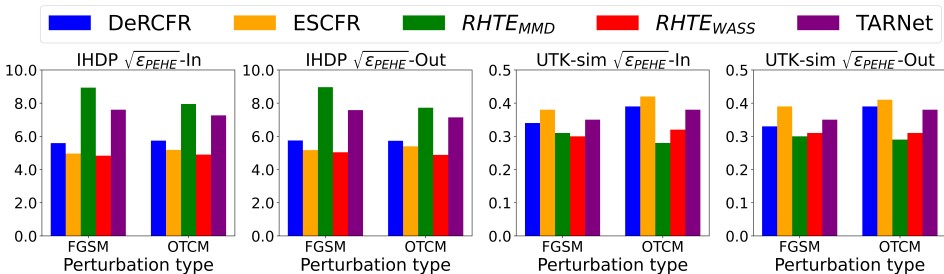

Figure 1: Performance comparison between the models under different types of perturbation on the IHDP and Image datasets.

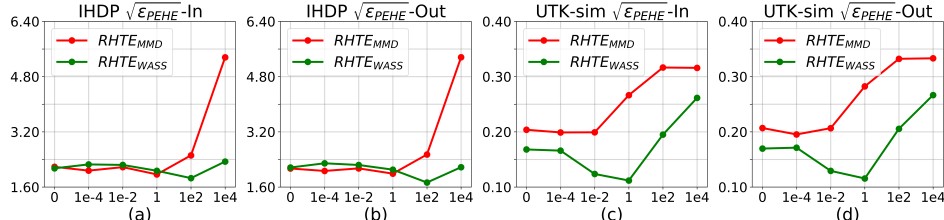

Figure 2: Influence of the weight parameter $\beta$. (a-b) present the performance on the IHDP dataset, while (c-d) show the result of the UTK-sim dataset.

This observation suggests that our framework can indeed improve the model's robustness even if the input points have been perturbed. For our framework, the strategies of Lipschitz regularization and RKHS regularization seem to have different advantages under different settings, and they alternatively achieve the best performances, which is analogous to the results observed in Table 1. Based on this observation, we speculate that, for Multimodal datasets, the $\text{RHTE}_{\text{MMD}}$ method can be leveraged to build a more robust treatment effect model. Otherwise, the $\text{RHTE}_{\text{WASS}}$ may also be competitive.

**Ablation Study on $\beta$.** After evaluating our model as a whole, we would like to study whether different designs in our model are necessary. In order to answer this question and illustrate the influence of the proposed terms, in this section, we conduct the ablation study, where the hyper-parameters settings follow the above experiments and we compare our model by varying the regularization penalty $\beta$. For optimization objective (7), the regularization influence will decrease when the regularization penalty $\beta$ becomes smaller. We tune $\beta$ in [0,1e-4,1e-2,1,1e2,1e4]. The results are presented in Figure 2. We can see that the best performance is usually achieved when $\beta$ is 1. This agrees with our opinion in section 3, i.e., too small $\beta$ may introduce too imbalance representation into the training process, while too large $\beta$ may severely impact the predictions made by the treatment effect model. The results indeed prove the proposed regularizer's effectiveness. By tuning $\beta$ in proper ranges, we are allowed to achieve better trade-offs to improve the treatment effects estimation performance.

## 6 CONCLUSION

By noting that previous representation learning methods seldom deal with measurement error in causal inference, especially covariate perturbation, which is of great significance in real-world study, we propose an RHTE framework to make robust CATE estimation under covariate perturbation. The estimator is derived by controlling empirical loss, Lipschitz constant, and representation distribution simultaneously. Generalization bounds on different types of adversarial losses are derived, implying the robustness of the RHTE estimator from a theoretical point of view. Experiments on various datasets are finally conducted to manifest the strong and robust performance of RHTE under different settings. This article opens a new perspective on the understanding of covariate perturbation through adversarial learning and enables representation learning methods to cope with covariate perturbation, which greatly broadens its application scenarios. A possible shortcoming is that this paper considers spherical perturbation on covariates. Dealing with more comprehensive types of perturbation, and dealing with the case when perturbation occurs not only on covariates but also on treatments and outcomes are interesting topics to be discussed in future research.

Table 2: Performance comparison between the models training in the way of adversarial defense-based methods.

| Datasets | IHDP | | UTK-sim | |
|---|---|---|---|---|
| Task | In-sample | Out-sample | In-sample | Out-sample |
| TARNet (T) | $2.32 \pm 3.13$ | $2.80 \pm 4.08$ | $0.25 \pm 0.02$ | $0.26 \pm 0.03$ |
| TARNet (A) | $2.10 \pm 2.66$ | $2.10 \pm 2.63$ | $0.12 \pm 0.01$ | $0.12 \pm 0.01$ |
| TARNet (D) | $2.29 \pm 2.78$ | $2.24 \pm 2.49$ | $0.13 \pm 0.01$ | $0.14 \pm 0.01$ |
| DeRCFR (T) | $2.92 \pm 3.94$ | $3.35 \pm 4.83$ | $0.30 \pm 0.03$ | $0.30 \pm 0.03$ |
| DeRCFR (A) | $2.83 \pm 4.34$ | $2.86 \pm 4.42$ | $0.12 \pm 0.01$ | $0.14 \pm 0.02$ |
| DeRCFR (D) | $2.92 \pm 3.82$ | $2.87 \pm 3.63$ | $0.18 \pm 0.02$ | $0.19 \pm 0.03$ |
| ESCFR (T) | $3.21 \pm 4.60$ | $3.78 \pm 5.54$ | $0.30 \pm 0.01$ | $0.30 \pm 0.02$ |
| ESCFR (A) | $3.70 \pm 5.17$ | $3.95 \pm 5.53$ | $0.18 \pm 0.02$ | $0.19 \pm 0.02$ |
| ESCFR (D) | $3.36 \pm 4.86$ | $3.55 \pm 5.03$ | $0.22 \pm 0.01$ | $0.24 \pm 0.02$ |
| $RHTE_{MMD}$ | $\mathbf{1.97 \pm 2.66}$ | $\mathbf{1.99 \pm 2.54}$ | $0.27 \pm 0.05$ | $0.28 \pm 0.06$ |
| $RHTE_{WASS}$ | $2.13 \pm 2.94$ | $2.12 \pm 2.79$ | $\mathbf{0.11 \pm 0.01}$ | $\mathbf{0.11 \pm 0.01}$ |

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

## A    EXTRA DEFINITIONS

In this section, we propose or recall the following definitions, which will be used in the proof.

**Definition 3.** Let $\Phi : \mathcal{X} \rightarrow \mathcal{R}$ be a representation function, $f : \mathcal{R} \times \{0, 1\} \rightarrow \mathcal{Y}$ be a hypothesis predicting the outcome of a unit's features $x$ given the representation covariates $\Phi(x)$ and the treatment assignment $t$. Let $L : \mathcal{Y} \times \mathcal{Y} \rightarrow \mathbb{R}_+$ be a loss function. The expected adversarial factual and counterfactual losses of $\Phi$ and $f$ are:

$$\epsilon_{Fadv}(f, \Phi) = \int_{\mathcal{X} \times \mathcal{T} \times \mathcal{Y}} L(y, f(\Phi(x_{adv}), t)) p(x, t, y) dx dt dy,$$

$$\epsilon_{CFadv}(f, \Phi) = \int_{\mathcal{X} \times \mathcal{T} \times \mathcal{Y}} L(y, f(\Phi(x_{adv}), t)) p(x, 1 - t, y) dx dt dy. \qquad (15)$$

**Definition 4.** The expected adversarial factual treated and control losses are:

$$\epsilon_{Fadv}^{t=1}(f, \Phi) = \int_{\mathcal{X} \times \mathcal{Y}} L(y, f(\Phi(x_{adv}), 1)) p(x, y | T = 1) dx dy,$$

$$\epsilon_{Fadv}^{t=0}(f, \Phi) = \int_{\mathcal{X} \times \mathcal{Y}} L(y, f(\Phi(x_{adv}), 0)) p(x, y | T = 0) dx dy. \qquad (16)$$

Accordingly, we can obtain an immediate results $\epsilon_{Fadv}(f, \Phi) = P(t = 1) \epsilon_{Fadv}^{t=1}(f, \Phi) + P(t = 0) \epsilon_{Fadv}^{t=0}(f, \Phi)$.

**Definition 5.** The estimation of treatment effect by an hypothesis $f$ and a representation function $\Phi$ for unit $x$ is:

$$\hat{\tau}(x) = f(\Phi(x), 1) - f(\Phi(x), 0), \qquad (17)$$

**Definition 6.** The expected Precision in Estimation of Heterogeneous Effect (PEHE) (Hill, 2011) loss of $f$ and $\Phi$ is:

$$\epsilon_{PEHE}(f) = \int_{\mathcal{X}} (\hat{\tau}(x) - \tau(x))^2 p(x) dx. \qquad (18)$$

and its adversarial version is defined as

$$\epsilon_{PEHEadv}(f) = \int_{\mathcal{X}} (\hat{\tau}(x_{adv}) - \tau(x_{adv}))^2 p(x) dx. \qquad (19)$$

**Definition 7.** Integral Probability Metric (IPM). For two probability density functions $p, q$ defined over $\mathcal{S} \in \mathbb{R}^d$, and for a function family $G$ of functions $g : \mathcal{S} \rightarrow \mathbb{R}$, The IPM is (Shalit et al., 2017):

$$\text{IPM}_G(p, q) := \sup_{g \in G} \left| \int_{\mathcal{S}} g(s)(p(s) - q(s)) ds \right|. \qquad (20)$$

## B    PROOF OF THEOREM 1

*Proof.* We reformulate the expected factual loss of $\Phi$ and $f$ as:

$$\epsilon_F(f, \Phi) = \mathbb{E}_{(x, t, y) \sim \mathcal{D}} [L(y, f(\Phi(x)), t)]$$

and its empirical factual loss is:

$$\hat{\epsilon}_F(f, \Phi) = \frac{1}{m} \sum_{i=1}^{m} L(y_i, f(\Phi(x_i), t_i))$$

For $t = 0, 1$, We can partition $\mathcal{D}_t$ into $2\mathcal{N}(\epsilon/2, \mathcal{X}, || \cdot ||_{\mathcal{X}}) \times \mathcal{N}(\epsilon/2, \mathcal{Y}, || \cdot ||_{\mathcal{Y}})$ subsets where $\mathcal{N}(\epsilon/2, \mathcal{X}, || \cdot ||_{\mathcal{X}})$ is the $\epsilon/2$-covering number of $\mathcal{X}$ and $\mathcal{N}(\epsilon/2, \mathcal{Y}, || \cdot ||_{\mathcal{Y}})$ is the $\epsilon/2$-covering number of $\mathcal{Y}$. For two samples $x_1$ and $x_2$ who belong to a same subset, we have $||x_1 - x_2||_{\mathcal{X}} \leq \epsilon$, and the corresponding outcomes $y_1$ and $y_2$ satisfies $||y_1 - y_2||_{\mathcal{Y}} \leq \epsilon$. Since $\mathcal{X}$ and $\mathcal{Y}$ are both bounded sets in $\mathcal{R}^d$ and $\mathcal{R}$, respectively, from finite covering theorem, $\mathcal{D}_t$ can be covered by finite open sets. We have the following lemma:

**Lemma 2.** *Let $K_t$ be the covering number of $\mathcal{D}_t$ using $\epsilon$-balls under metric $|| \cdot ||$ and $\{\mathcal{D}_1^t, ..., \mathcal{D}_{K_t}^t\}$ be the partitioned subsets of $\mathcal{D}_t$ as defined above, and $K = K_1 + K_2$. Recall that $D = \{(x_i, t_i, y_i)\}_{i=1}^m$ is the observational data. Let $N_i^t$ be the number of observations from $D$ that fall into $\mathcal{D}_i^t$. Note that $\{|N_1^1|, ..., |N_{K_1}^1|, |N_1^2|, ..., |N_{K_2}^2|\}$ is an IID multinomial random variable with parameters $m$*

and $\{\mu(\mathcal{D}_1^1), ..., \mu(\mathcal{D}_{K_1}^1), \mu(\mathcal{D}_1^2), ..., \mu(\mathcal{D}_{K_2}^2)\}$. *By the Breteganolle-Huber-Carol inequality (Xu & Mannor, 2012), the following holds with probability at least* $1 - \delta$:

$$\sum_{t=1}^{2} \sum_{i=1}^{K_t} \left| \frac{|N_i^t|}{m} - \mu(\mathcal{D}_i^t) \right| \leq \sqrt{\frac{2K \ln 2 + 2 \ln(1/\delta)}{m}}$$

From the lemma we have
$$|\epsilon_F(f, \Phi) - \hat{\epsilon}_F(f, \Phi)|$$

$$= \left| \sum_{t=1}^{2} \sum_{i=1}^{K_t} \mathbb{E}\left[ L(y, f(\Phi(x), t)) | (x, t, y) \in \mathcal{D}_i^t \right] \mu(\mathcal{D}_i^t) - \frac{1}{m} \sum_{i=1}^{m} L(y_i, f(\Phi(x), t_i)) \right|$$

$$\leq \left| \sum_{t=1}^{2} \sum_{i=1}^{K_t} \mathbb{E}\left[ L(y, f(\Phi(x), t)) | (x, t, y) \in \mathcal{D}_i^t \right] \frac{|N_i^t|}{m} - \frac{1}{m} \sum_{i=1}^{m} L(y_i, f(\Phi(x), t_i)) \right|$$

$$+ \left| \sum_{t=1}^{2} \sum_{i=1}^{K_t} \mathbb{E}\left[ L(y, f(\Phi(x), t)) | (x, t, y) \in \mathcal{D}_i^t \right] \mu(\mathcal{D}_i^t) - \sum_{t=1}^{2} \sum_{i=1}^{K_t} \mathbb{E}\left[ L(y, f(\Phi(x), t)) | (x, t, y) \in \mathcal{D}_i^t \right] \frac{|N_i^t|}{m} \right|$$

$$\leq \left| \frac{1}{m} \sum_{t=1}^{2} \sum_{i=1}^{K_t} \sum_{j \in N_i^t} \max_{(x,t,y) \in \mathcal{D}_i^t} |L(y_j, f(\Phi(x_j), t_j)) - L(y, f(\Phi(x), t))| \right|$$

$$+ \left| \max_{(x,t,y) \in \mathcal{D}} |L(y, f(\Phi(x), t))| \sum_{t=1}^{2} \sum_{i=1}^{K_t} \left| \frac{|N_i^t|}{m} - \mu(\mathcal{D}_i^t) \right| \right|$$

$$\leq \lambda_l \Lambda_f \epsilon + \mathcal{C}_d \sum_{t=1}^{2} \sum_{i=1}^{K_t} \left| \frac{|N_i^t|}{m} - \mu(\mathcal{D}_i^t) \right|$$

By combining Lemma 2 and apply it in Lemma 1, the proof of Theorem 1 is done.

$\square$

## C    PROOF OF THEOREM 2

*Proof.* Following the proof of generalization bound on PEHE in Theorem 1 of Shalit et al. (2017) by substituting all $x$ apart from those in probability functions with its adversarial sample $x_{adv}$, we have

$$\epsilon_{PEHEadv}(f, \Phi) \leq 2(\epsilon_{CFadv}(f, \Phi) + \epsilon_{Fadv}(f, \Phi) - C_Y)$$
$$\leq 2\left( \epsilon_{Fadv}^{t=0}(f, \Phi) + \epsilon_{Fadv}^{t=1}(f, \Phi) \right) + 2\left( C_\Phi \cdot IPM_G(p_\Phi^{t=1}, p_\Phi^{t=0}) - C_Y \right),$$

Combining it with Theorem 1 gets the result. $\square$

## D    EXPERIMENTAL DETAILS

We implement our methods based on BNN (Johansson et al., 2016). We use the same set of hyperparameters for RHTE across three datasets. More specifically, we adopt 3 fully-connected exponential-linear layers for the representation function $\Phi$ and 3 similar architecture layers for the treatment effect prediction function $f$. The difference is that layer sizes are 200 for the former, and 100 for the latter. Batch normalization (Ioffe & Szegedy, 2015) is applied to facilitate training, and all but the output layer use ReLU (Rectified Linear Unit) (Agarap, 2018) as activation functions whose Lipschitz constant is less than or equal to 1. Additionally, we set batch size to 64 and learning rate to 0.01 with 0.0001 weight decay. In the main optimization objective, we set $\alpha$ and $\beta$ both to 1. Following the common settings (Shalit et al., 2017; Wu et al., 2022; Wang et al., 2023), we present within-sample and out-of-sample results that are calculated on the training and test set respectively. For the ACIC dataset, we conduct experiments over randomly picked 100 realizations with 63/27/10 train/validation/test splits. For IHDP dataset, following the common settings in Qin et al. (2021); Shalit et al. (2017), we average over 100 replications of the outcomes with 63/27/10 train/validation/test splits. For the UTK-sim dataset, we average over 10 replications of the outcomes

---

**Algorithm 1** Learning algorithm of our model

---

Indicate the observational data $(x_1, t_1, y_1), ..., (x_m, t_m, y_m)$;
Indicate the scaling parameter $\alpha$ and $\beta$;
Initialize all the model parameters;
Indicate the epoch number $E$;
Compute $u = \frac{1}{m} \sum_{i=1}^{m} t_i$;
Compute $w_i = \frac{t_i}{2u} + \frac{1-t_i}{2(1-u)}$ for $i = 1, ..., m$;
**for** $e = 0$ **to** $E$ **do**
    Sample mini-batch data $\mathcal{B}$ from $\mathcal{D}$;
    Compute the gradients of the regularization:

$$g_1 = \nabla_W \beta \mathcal{R}(f)$$

    Compute the gradients of the IPM term:

$$g_2 = \nabla_W \alpha IPM_G(\hat{p}_\Phi^{t=1}, \hat{p}_\Phi^{t=0})$$

    Compute the gradients of the empirical loss:

$$g_3 = \nabla_W \frac{1}{|\mathcal{B}|} \sum_{i=1}^{|\mathcal{B}|} w_i L(y_i, f(\Phi(x_i), t_i))$$

    Obtain the step size scalar $\eta$ with the Adam;
    Update the parameters:

$$W \leftarrow W - \eta(g_1 + g_2 + g_3)$$

**end for**

---

with 63/27/10 train/validation/test splits. For the TC-sim dataset, we average over 3 replications of the outcomes with 63/27/10 train/validation/test splits.

## E SUPPLEMENTARY AND ALGORITHM

Supplementary material includes dataset links, source codes, and the guidelines for running experiments. We present our CATE estimation algorithm in Algorithm 1

