# OpenReview forum: "Robust Heterogeneous Treatment Effect Estimation under Covariate Perturbation"
_ICLR.cc/2025/Conference — Submitted to ICLR 2025_

### Official Review · Reviewer_pjWR · 2024-10-18

**Soundness:** 2
**Presentation:** 1
**Contribution:** 2
**Rating:** 5
**Confidence:** 3

**Summary:**

The methodology outlined in the paper focuses on addressing covariate perturbation—essentially measurement errors in covariates—in Conditional Average Treatment Effect (CATE) estimation. The approach involves using adversarial learning techniques to enhance the robustness of treatment effect estimations under such perturbations. Theoretical guarantees are provided in terms of PEHE (Precision in Estimation of Heterogeneous Effect) loss, ensuring that these methods perform reliably even when the covariate observations are perturbed. They evaluate the effectiveness of their proposed CATE estimation methods using two standard benchmark datasets—ACIC and IHDP—and two synthetic datasets, UTK-sim and TC-sim.

**Strengths:**

- The paper introduces novel regularizations—Orthonormality Regularization and RKHS Regularization—to control the Lipschitz constant of the causal prediction model.
- Their theoretical framework provides generalization bounds showing that the model's error on adversarial samples can be controlled.

**Weaknesses:**

- The paper does not sufficiently discuss the practical relevance of adversarial perturbations in treatment effect estimation. It's unclear if real-world scenarios would commonly present such adversarial challenges, which raises questions about the relevance of the proposed approach.
- The specific choice of uniform noise for the perturbations  in the experiments  may be tailored to showcase the strengths of the proposed method. Unfortunately, the paper is lacking ablations for different noise types.
-  While the authors provide details about the hyperparameters used, these settings are fixed throughout all experiments, and no hyperparameter search is conducted. This lack of finetuning could limit the reproducibility of the findings, as the performance of the baselines  might vary with different hyperparameter configurations.

**Questions:**

- How does the proposed method perform when the covariates are perturbed with different noises, eg gaussian or exponential?
- Why are the hyperparameters of the baselines never tuned?
- Can the author provide some real examples where covariates are perturbed in a treatment effect estimation setting?

---

### Official Review · Reviewer_GqhY · 2024-11-02

**Soundness:** 3
**Presentation:** 2
**Contribution:** 2
**Rating:** 3
**Confidence:** 5

**Summary:**

This work addresses the covariate perturbation problem for CATE estimation. Building on the insights that the worst-case PEHE under adversarial perturbation can be effectively controlled through restricting the Lipschitz constant of the potential outcome predictors, this work proposes a representation-learning framework Robust HTE (RHTE). In particular, the authors propose two types of regularizations to control the Lipschitz constant - the Orthonormality Regularization and RKHS Regularization. On top of the theoretical results to justify their framework, the authors also provide empirical studies to corroborate RHTE's efficacy.

**Strengths:**

- Robust CATE is an important problem to investigate.
- The proposed method is intuitive and reasonable.
- The experiments appear comprehensive (though with some aspects needing clarification)

**Weaknesses:**

## Technical (Framework) Novelty
- The proposed framework is simply the well-known balancing framework for CATE [1] with the an additional regularization of the predictor's Lipschitz constant. Personally I think this is not a big problem.
- Notably, the two "proposed" regularization methods are also well-known results: the techniques of upper bounding the Lipschitz constant with spectral norm are already widely used [2,3]; the RKHS result is a standard result in kernel-based learning.

## Theoretical Novelty
- It is more than intuitive that controlling the Lipschitz constant of the predictor can help control the adversarial loss because it represents the maximum change of prediction one adversarial perturbation can cause. However, note that by controlling the Lipchitz constant of the predictor, you are also making an **implicit assumption** that the true potential outcome function has a Lipchitz constant **on the same scale**. **In other words, the EPHE can be arbitrarily bad if you are controlling the Lipschitz constant of your predictor yet the true potential outcome function has large-enough Lipchitz constant**. Please ruminate Theorem 2 and you should see this implicit assumption. This needs to be discussed.
- Lemma 1 is a fairly obvious result. But **I believe it is fine since it only serves to motivate your framework**.
- The theoretical results (Theorem 1 and 2) appear **disconnected and unnecessary** to the framework. See more details below.
- Theorem 1 is just an empirical process result based on the covering number argument to upper bound the true factual (adversary) loss with the empirical one and a finite-sample gap. And, to the best of my knowledge, leaving theoretical results with raw covering number **does not convey any insights** because we usually continue to give instance-specific upper bounds for the covering number. The fact that the empirical process on compact space (i.e., with bounded covering number) has \sqrt{1/n} concentration is standard in the community.
- To continue on the last comment, the remark after Theorem 1 that "Theorem 1 ... rationalizes the control on Lipschitz constant" has nothing to do Theorem 1. **You can already reach this conclusion with Lemma 1**. The finite-sample concentration result in Theorem 1 has nothing to do with it.
- Theorem 2 is a simple (if not entirely trivial) extension of the results in [1]. **I don't think it justifies to be an original theorem in the main text.** Just writing it in plain language and referring [1] should be enough. Please consider editing it.


[1] Uri Shalit, Fredrik D Johansson, and David Sontag. Estimating individual treatment effect: general-
ization bounds and algorithms. In International Conference on Machine Learning, pp. 3076–3085.
PMLR, 2017.
[2] https://towardsdatascience.com/lipschitz-continuity-and-spectral-normalization-b03b36066b0d
[3] K. Kurach, M. Lucic, X. Zhai, M. Michalski, and S. Gelly. A Large-Scale Study on Regularization and Normalization in GANs (2019), ICML 2019.

**Questions:**

## Comparison with Existing Methods
In the introduction, the authors comment on the existing approaches that "these
assumptions cannot be tested from observed data, and restrict the generality of the methods". However, if my understanding is correct, the assumptions in this framework are also **very difficult to verify**. For example, how to verify the "spherical perturbation" assumption and the Lipschitzness of the potential outcome functions?

## Technical Questions
The experiment section is poorly written and needs some clarification.
- There are two regularizations. Which one is used in the experiments?
- I think it is better to illustrate the meaning of "in-sample" and "out-sample" in your manuscript.
- In Table 1, why the performance of "in-sample" is also significantly improved?
- The experiment setting of "Robustness Comparison" is unclear. Are you directly applying 3 adversarial ML methods for CATE estimation? How exactly is this performed?
- How do you choose the hyperparameter \beta? The authors state at the end of Section that "by tuning \beta in proper ranges, we are allowed to achieve better trade-offs to improve the treatment effects estimation performance". However, without counterfactual outcomes in real-world applications, based on what criterion can you tune this hyperparameter?

## Minors
1. (p-1)!! on line 185 is undefined
2. x_i, t_i, y_i in Equation (7) was undefined later in Section 4.3 yet their first appearance is in Section 4.2.
3. The left ' in 'X(T)', 'X(A)', 'X(D)' are flipped.
4. All the specific section references, e.g., Section 3, should be capitalized.
5. There are quite some typos in Section 5. I recommend the authors to edit it carefully during rebuttal.

---

### Official Review · Reviewer_no77 · 2024-11-03

**Soundness:** 2
**Presentation:** 3
**Contribution:** 2
**Rating:** 5
**Confidence:** 3

**Summary:**

The paper presents a robust method for training CATE estimators by introducing an additional regularizer to control the model's Lipschitz continuity. The authors provide theoretical performance results and evaluate their method across several datasets.

**Strengths:**

The paper studies an interesting problem, i.e, the robustness of CATE estimation models to distribution shift. It presents few interesting theoretical results and presents good empirical evaluation.

**Weaknesses:**

My main concern with the paper is its incremental nature. While the results are interesting, they are a direct consequence of prior results in adversarial settings for general learning scenarios. Moreover, the paper lacks motivation on why the adversarial setting is particularly relevant for causal inference—specifically, how it differs from the general learning problem. For instance, can a model be robust to adversarial attacks over the factual distribution but not over its counterfactual distribution?

**Questions:**

1. How do the robustness measure interact with the violation of the assumption, especially the conditional unconfoundedness assumption?

2. Is there an informative way to know when to use which one of the presented regularizer (Orthonormality Regularization and RKHS Regularization). And under what conditions might one be preferred over the other? Since in practice only the factual distribution, I understand that this can be challenging.

3. Can you provide details on hyperparmeters tuning and if you have any insights on how to do this in real world settings when no counterfactual samples are available to validate?

4. Which of the two presented approaches was used in practice (Orthonormality Regularization and RKHS Regularization)?

5. While the bound in theorem 2 is interesting, how to guarantee that all three terms can converge together? Are there any possible experiments to conduct over simple data to see how the convergences of the three terms are interacting?

6. Are there potential scenarios where this method might fail? For example, if the true underlying function is not Lipschitz for any order, would this method still risk significant degradation in performance?

---

### Official Review · Reviewer_5mK7 · 2024-11-04

**Soundness:** 3
**Presentation:** 2
**Contribution:** 3
**Rating:** 5
**Confidence:** 2

**Summary:**

The paper introduces the RHTE framework, for estimating heterogeneous treatment effects under covariate perturbations. The main contribution is a novel representation learning approach that enhances the robustness of treatment effect estimations by incorporating adversarial machine learning techniques, with a specific focus on handling covariate perturbations.

**Strengths:**

- The framework is well-supported by theoretical derivations, including generalization bounds for adversarial PEHE loss.
- Experiments on both synthetic and real-world datasets are conducted, demonstrating the model's robustness and effectiveness.

**Weaknesses:**

- The paper only considers spherical perturbations, which may limit the method's applicability to broader real-world perturbations.
- The use of adversarial training and Lipschitz regularization may complicate implementation, which could be a barrier for practitioners without expertise in adversarial machine learning.

**Questions:**

- How does the choice of spherical perturbations impact the robustness of the RHTE framework? Could other forms of perturbations (e.g., elliptical or non-uniform) be more representative of real-world scenarios?
- What are the limitations of using Integral Probability Metric (IPM) in this context? Would other distance metrics perform differently in aligning treated and control representations?
- What insights can be drawn about the model’s performance in balancing the treated and control distributions? Does the RHTE consistently outperform traditional models, even under extreme perturbations?

---

### Meta-Review · Area_Chair_VtPq · 2024-12-22

**Metareview:**

This paper is at the intersection of causal inference and adversarial robustness and develops a robust representation learning algorithm by applying adversarial learning.  The main concern was the incremental nature of the paper because it felt like a directly from prior results from adversarial training and existing results from RKHS. There were also questions on the types of perturbations to be used.

**Additional Comments On Reviewer Discussion:**

There was no author reply and all reviewers are in agreement for rejection.

---

### Decision · Program_Chairs · 2025-01-22

Reject